# Surface Defects Improved SERS Activity of Nanoporous Gold Prepared by Electrochemical Dealloying

**DOI:** 10.3390/nano13010187

**Published:** 2022-12-31

**Authors:** Ling Zhang, Zhiyu Jing, Zhexiao Li, Takeshi Fujita

**Affiliations:** 1School of Optical-Electrical and Computer Engineering, University of Shanghai for Science and Technology, Shanghai 200093, China; 2School of Environmental Science and Engineering, Kochi University of Technology, Kochi 782-8502, Japan

**Keywords:** nanoporous gold, electrochemical dealloying, surface morphology, chemical state, surface-enhanced Raman scattering

## Abstract

Nanoporous metals possess excellent catalytic and optical properties that are related with surface morphology. Here, we modulated the ligament surface of nanoporous gold (NPG) by controlling electrochemical dealloying and obtained NPG with an improved enhancement of its surface-enhanced Raman scattering (SERS) property. We found that both high-density atomic steps and kinks on the curved surfaces and high-content silver atoms close to the ligament surface contributed to the high SERS ability. The presented strategy will be useful for the fabrication of nanoporous metal with an excellent surface that is needed for sensing, conversion, and catalytic.

## 1. Introduction

Nanoporous metals (NPM) are a kind of nanostructured material with bicontinuous ligaments and channels on a nanosize scale. They are usually prepared by selective dissolving of active component from a compound precursor [1,2,3,4,5]. The nanoporous metals exhibit excellent properties and potential applications, such as catalysis; energy conversion and storage; and biomedical sensing and optical enhancement [6,7,8]. Optical properties of nanostructured metals are dramatically different from their bulk counterparts due to surface plasmon resonance (SPR) [8,9,10,11]. For the NPMs with ligaments/channels smaller than the wavelength of visible light, significant SPR can be excited within the visible wavelength, and the resonance wavelength region tuned by controlling ligaments/channels sizes.

Nanoporous gold (NPG) was proven to be a selectable SPR generator and applied in surface-enhanced Raman scattering (SERS) [12,13] due to its high biocompatibility, good chemical stability, easy surface modification via thiol-gold linker chemistry, and compatibility with conventional microfabrication [7,8]. Several methods reported in the literatures for fabricating NPG-based SERS substrate [13,14,15,16]. According to the previous results, hierarchical structure; ligament size and shape; and component and distance between nearby ligaments affect the SERS ability of NPM. Qian et al. reported one order SERS improvement by mechanically ruptured nanoprous gold (NPG) to form fracture surfaces [17]. Lang et al. obtained strong electromagnetic fields by reducing the size of the nanopore to increase the coupling effect between neighbor gold ligaments [18]. Chen et al. obtained better SRES enhancement by developing a silver-coated nanoporous nanostructure [19]. Zhang et al. reported wrinkled nanoporous gold that contains abundant Raman-active nanogaps and yields ultrahigh SERS [20]. All mentioned approaches require secondary treatment to either modify the surface ligament or tune the morphology of the NPG films. Here, we used the electrochemical dealloying method to successfully construct more hot spots on the ligament surface of NPG, resulting in obvious SERS improvement.

Dealloying, selective dissolution of less noble elements from an alloy, enables the preparation NPMs with a characteristic size tuned from a few nanometers to several microns. Electrochemical dealloying (Ele-dealloying) can tune the length scale of ligaments/channels in NPMs more precisely by controlling etching time and dealloying potential [3,8]. Due to the mechanism of Ele-dealloying, dealloying thresholds and critical potentials are two key parameters in characterizing dealloying, both of which affect the surface diffusion and formability of the nanoporous structure [21]. In this work, we systematically investigated the influence of applied potential on the surface morphology of NPG, and found that adopting a constant potential slightly higher than the critical potential to perform Ele-dealloying is possible to generate more active sites on the ligament surface, including kinks; step edges and vacancies; and more residual silver closed to the surface during the formation of porosity. Due to the high density of surface defects, the surface area to volume ratio of NPG prepared with Ele-dealloying is larger than that of free-dealloyed NPG with identical ligament size and silver content, meanwhile, more than 20 times improvement of the SERS effect achieved from NPG prepared with Ele-dealloying. Therefore, electrochemical-dealloyed NPG is a good candidate for SERS and other surface-dependent applications [22]. 

## 2. Materials and Methods

NPG films (100 nm thick) were prepared by Ele-dealloying with a standard three electrodes configuration: Ag-Au thin alloy films as working electrode, a platinum sheet as counter electrode and Ag/AgCl electrode as reference. The electrolyte was a 2 M aqueous solution of nitric acid. NPG control samples were prepared by free dealloying in 70 vol % nitric acid. All chemicals were analytical grade and used without further purification. The as-prepared NPG films were carefully rinsed with distilled water (18.2 MΩ·cm) to remove the remaining nitric acid, and then dried in the air for further measurement.

Microstructure characterization was investigated by scanning electron microscope (SEM) (JEOL, JIB-4600F), and transmission electron microscope (JEOL JEM-2100F.X-ray photoelectron spectroscopy (XPS AxIS-ULTRA-DLD) was employed to investigate the surface composition and chemical states. The characteristic length scales of NPG, defined by the equivalent diameters of nanopores or the gold ligaments, are measured from statistical analysis of digital SEM images by a rotational fast Fourier transform method [23,24]. Surface atomic structure of NPG was characterized using spherical-aberration-corrected TEM (Cs-corrected TEM) in a scanning TEM (STEM) mode with a high-angle annular dark-field (HAADF) detector.

A micro-Raman spectrometer (Renishaw InVia RM 1000) with an excitation laser wavelength of 514.5 nm was used for SERS measurements. The laser power was set at a low value of 0.1 mW to avoid possible laser irradiation damage of the probe molecules. A 10^−6^ M Rhodamine 6G(R6G) aqueous solution was prepared as probe molecules. As-prepared NPG films were soaked in 10^−6^ M R6G aqueous solution for two hours to modify molecules on the ligaments, and then washed with distilled water and dried for SERS measurement. 

## 3. Results and Discussion

NPG films were prepared by potentiostatic dealloying. In order to decide etching potential, cyclic voltammogram of Ag_65_Au_35_ (at. %) was measured (Figure 1a) in 2 M nitric acid aqueous solution. As shown in Figure 1a, the critical potential is around 1.0 V. Thus, three voltages higher than 1.0 V (1.05 V, 1.10 V, and 1.15 V) were applied to fabricate NPG films, and the corresponding current variation during dealloying is shown in Figure 1b. The initial current at the applied potential of 1.05 V is significantly smaller than that at 1.10 V and 1.15 V. At 1.05 V, a current platform appears as dealloying time of 0~1500 s in the relative steady state. Due the reduction of silver elements, with the extension of dealloying time, the current intensity decreased smoothly, resulting in the coarsening of ligament [23]. However, no current platform arose when a higher potential of 1.10 V and 1.15 V was applied. A large current was generated since the beginning and thereupon dropped sharply, indicating local passivation rapidly formed due to the high dissolving speed of Ag atoms and fast diffusion speed of Au adatoms [3]. Accordingly, dealloying time dependences of the characteristic length (ligament and pore sizes) appear opposite trends (Figure 1c). The ligament and pore size are estimated using a rotational fast Fourier transform method from the SEM images of NPG films [24,25]. The ligament and pore sizes are identical with Ag_65_Au_35_ (at. %) as the precusor. At 1.05 V, the ligament and pore size increased slower at the initial stage and faster latterly; at 1.15 V the ligament and pore size increased faster at the beginning and slower down lately. The increasing slope of NPG prepared at 1.15 V is comparable with free dealloy with highly concentrated nitric acid. It follows that applied adaptive voltage can effectively control the dealloying speed, such that it is possible to modify the surface morphology of NPG which is mainly determined by percolation dissolution for the alloy and terrace dissolution on the surface [26,27]. 

SERS ability of nanostructured metals depends on the shape, size, content, and surface morphology. We checked the SERS intensities of molecules adsorbed on the NPG prepared by Ele-dealloying and free dealloying. Figure 2a shows the pore size dependences of the Raman intensities. In the detected size region, the Raman intensity of free-dealloyed NPG deceased with increasing of ligament size, so as NPG prepared by electrochemical dealloying at 1.05 V. With identical ligament size, NPG prepared with electrochemical dealloying at 1.05 V exhibits much better SERS enhancement compared to others, even with relatively less residual silver. While the Raman intensity of NPG prepared by dealloying at 1.15 V increased first and then decreased. According to the previous reports both the ligament size and silver content influence the SERS ability of NPG [28,29], and small size and high silver content are good for SERS enhancement [30]. However, the NPG prepared with short dealloying time at 1.15 V, which possesses small characteristic length and plenty residual silver does not show prominent SERS ability, implying that some unknown factors other than nanopore sizes and residual silver significantly affect the SERS effect of the Ele-dealloyed NPG. 

We thus chose three samples prepared at different conditions but with identical characteristic lengths and similar residual silver in order to investigate the influence of applied potential on NPG and it SERS performance. NPG-Free: free dealloying for 600 s with ligament size about 14 ± 2 nm and 18 at.% residual silver; NPG-1.05 V Ele: Ele-dealloying at 1.05 V for 500 s with ligament size about 15 ± 4 nm and 17 at.% residual silver; and NPG-1.15 V Ele: Ele-dealloying at 1.15 V for 100 s with ligament size about 12 ± 3 nm and 20 at.% residual silver. The SERS spectra and morphologies of the three samples are shown in Figure 2b–e. From the SEM images it can be seen that all three samples are uniform, and the SERS intensity (Figure 2b) that is obtained on the surface of NPG-1.05 V Ele is about eight times stronger than that obtained from NPG-Free, and four times stronger than that obtained from NPG-1.15 V Ele. Although the ligament size of NPG-1.05 V Ele is larger and the residual silver is less, it still displays better SERS performance compared to NPG-Free, and NPG-1.15 V Ele confirmed that some additional factors affect the monolithic property. 

Due to the fact that the SERS effect strongly depends on the surface mophology of plasmonic substrates, the atomic-scale structure of the three samples were further studied. Figure 3 shows the STEM images of representative areas from the three samples. The insets are the histograms of scattering intensity of each atomic column along arrow direction. The z-contrast is different either due to the silver content, or due to the number of atoms in the atomic column. Comparing the intensity profiles from upmost surface regions (marked with number 1), steps can be seen for all the three samples. Comparing the intensity profiles numbered with 2 of the three samples, a gradual decrease in intensity over several atomic columns was observed due to the kinks and steps on the surface, and more variation can be observed from NPG-1.05 V Ele. Addtionally, The variation of profile intensity in the region numbered with 3 from NPG-1.05 V Ele is due to cavities or silver aggregation near the surface. In such cases, more surface atoms with low coordination numbers exist on NPG-1.05 V Ele, leading to a larger surface area to volume ratio (SVR). 

Estimate the SVR of the three samples with the single-atom layer REDOX reactions [31,32], we confirmed that the SVR of NPG-1.15 V Ele is comparable to that of NPG-Free, while the SVR of NPG-1.05 V Ele is more than two times larger than that of NPG-Free (see Appendix A, more details are given in Appendix A). Thus, surface defects affect SVR obviously, and the present selected applied potential is a key factor to form high defect ligaments surface.

The surface composition and chemical states were further investigated by XPS technique, Figure 4 show the XPS spectra of selected NPG films prepared with different conditions. Au 4f peaks are located around 84.5 eV and 88.2 eV (Figure 4a) [33]. Meanwhile, two peaks attributed to Ag3d_3/2_ and Ag3d_5/2_ located around 368.4 and 374.4 eV (Figure 4b) [4], which confirms the presence of metallic Ag near the surface. Additionally, the peak intensity of Ag 3d of NPG-Free is obviously weaker than that of NPG-Ele, indicating that the Ag element closer to the surface of NPG-Ele is more than that of NPG-Free, and even the entire Ag concentration of them are comparable. Accordingly, the peak intensity of Au 4f of NPG-Ele is a little weaker than that of NPG-Free, and further confirmed electrical dealloying can maintain more silver elements near the surface. With larger surface area and high surface Ag concentration, NPG-1.05 V Ele exhibited much better SERS enhancement than NPG-Free. Similarly, NPG-1.15 V Ele also contain higher surface silver concentration, but the SERS improvement was not as obvious as NPG-1.05 V Ele even with a smaller ligament size. Further characterization of the chemical states found that surface oxidation is another reason causing the abnormal SERS performance of NPG-1.15Ele. As shown in Figure 4c, the wide peak can be fitted to two Gaussian peaks with binding energy of 531.2 eV and 532.5 eV (see Appendix A), which indicate existence of different kinds of O species. The peak with bonding energy of 531.2 eV is associated with surface lattice O, while the 532.5 eV is attributed to surface adsorbed O_2_ [4]. The intensity of surface lattice O from NPG-1.05Ele is the smallest. Whereas, the intensity of adsorbed O_2_ is the largest. That is to say, the applied voltage effectively inhibited oxidation of surface lattice, and meanwhile retained more low coordination active sites for molecule adsorption. Surface chemical states analysis further testified that the applied voltage helps to remain more Ag element near the surface (see Appendix A) during dealloying without affecting porous formation. With more active sites and residual silver atoms near the surface, NPG prepared with 1.05 V electrochemical dealloying always exhibit better SERS performance even with larger ligament size (see Appendix A). 

According to the above experimental results, we fabricated the NPG with more effective surface and achieved more 20 times SERS improvement by using Ag_80_Au_20_ as the precursor (see Appendix A). Several silver rich regions and clear terrace line can be observed on the ligaments, which predict better SERS enhancement. It is generally believed that SERS enhancement mechanisms are ascribed to electromagnetic field enhancement which comes from the localized surface plasmon resonances on rough metallic nanostructures and/or the chemical enhancement which arises from electronic structural changes from the SERS substrates to the analytes [22,34]. The increased surface defects provide more active sites for molecules to adsorb which can increase the amount of substance in the local region. Correspondingly, the total SERS intensity was improved.

## 4. Conclusions

In general, we fabricated NPG films with electrochemical dealloying and obtained NPG with larger sa/vol and more active sites. The as-prepared NPG exhibited much higher SERS enhancement compared to free-dealloyed NPG, and both surface morphology and chemical states contributed to the improved SERS ability. The results indicate that electrochemical dealloying is an effective way to modulate the surface state of nanoporous metals and is a prospective way to optimize nanostructured materials for SERS and interface-related applications.

## Figures and Tables

**Figure 1 nanomaterials-13-00187-f001:**
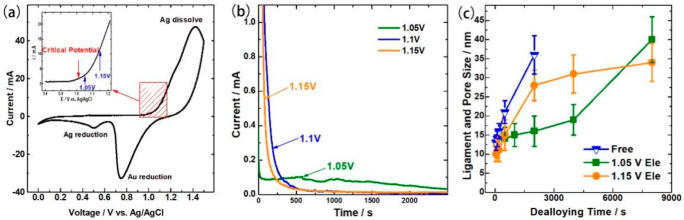
(**a**) Experimental cyclic voltammetry curve of Ag_65_Au_35_ in 2 M HNO_3_ acid solution (scan rate: 100 mVs^−1^). (**b**) Typical dealloying curves of Ag_65_Au_35_ in 2 M HNO_3_ at different potentials with the same mass and geometrical area. Insets are the enlarged curves of the selective region. (**c**) Dealloying time dependences of ligament and pore size.

**Figure 2 nanomaterials-13-00187-f002:**
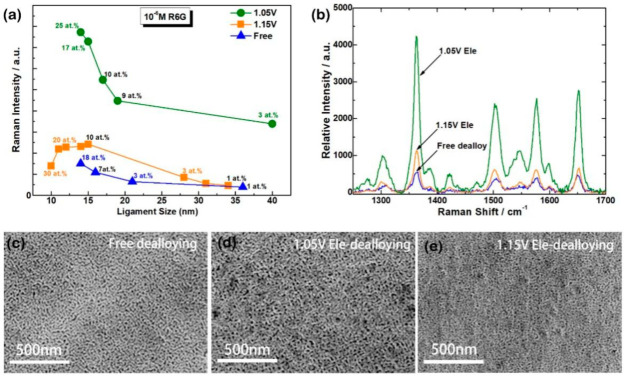
SEM images and SERS spectra of NPG with similar ligament size and residual silver that prepared at different conditions. (**a**) Pore size dependences of the Raman intensities, and the number near the data dots are the residual silver concentration after dealloying. (**b**) SERS spectra of selected three samples. SEM images of the three samples (**c**) NPG-Free, (**d**) NPG-1.05 V Ele, and (**e**) NPG-1.15 V Ele.

**Figure 3 nanomaterials-13-00187-f003:**
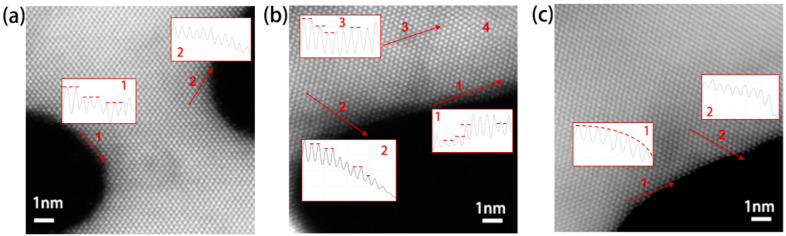
STEM images of the three selected samples. (**a**) NPG-Free, (**b**) NPG-1.05 V Ele, and (**c**) NPG-1.15 V Ele. The inset figures of (**a**–**c**) are the histograms of scattering intensity of each atomic column along arrow direction.

**Figure 4 nanomaterials-13-00187-f004:**
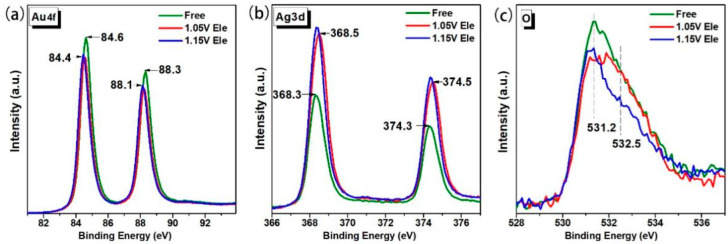
XPS spectra of selected NPG films prepared with different conditions. Binding energy of (**a**) Au 4f core levels, (**b**) Ag 3d core levels, and (**c**) O 1s level.

## Data Availability

All data concerning this study are contained in the present manuscript or in previous articles, whose references have been provided.

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
