# Peer review of "Surface Defects Improved SERS Activity of Nanoporous Gold Prepared by Electrochemical Dealloying"

_nanomaterials, 2022, doi:10.3390/nano13010187_

Round 1

Reviewer 1 Report

The most significant feature of this paper deals with simple and efficient SERS-enhanced NPG preparation technique proposed by authors. This electrochemical etching technique of Au-Ag alloy films and foils nearby their critical potentials of dealloying results in the nanoporous  materials with significantly enhanced SERS effect at their surface. The values of critical potentials of the different alloys can be easily obtained from the common CV measurements.

The efforts of authors to explain the observed phenomena using various physico-chemical measurements of these materials look less successful  and don't give sufficient scientific background for the observed effects. Taking into account these features, I would recommend to reduce significantly a discussion of the reasons and mechanism of the observed processes and to resubmit this paper as a short communication focused at the practically valuable aspects of the proposed NRG synthesis technique. The reasons of the observed effects and phenomena could be discussed further, after more detailed instrumental studies.

Reviewer 2 Report

The manuscript by Zhang and collaborators describes a systematic study of the electrochemical dealloying of gold-silver leaves to fabricate nanoporous structures. The authors carried out a routine study that has been investigated in many laboratories. In addition, the discussion lacks an in-depth explanation of the results. SERS results were not studied properly. References are very limited to self-citation and repeated authors. References should cover many labs and countries.

Overall, the manuscript must be rejected based on quality, presentation, and novelty.

Reviewer 3 Report

The manuscript focus on electrochemical dealloying by means of surface modulation of nanoporous gold materials.
The topic correlates to the journal.
The work has a clear structure.
All sections are required for a complete understanding.
Nevertheless, there are some minor issues that require to be addressed before proceeding with the publication, to enhance the quality and presentation to a broad audience.
The abstract reports a consistent summary of the article core, although it might be improved by a few additional, thorough sentences, to catch the readers’ attention.
Aim and hypothesis are missing and will strengthen the core research and manuscript novelty.
Please, better describe the details of the “Materials and methods” section: how are in detail produced the NPG films? Define the brand or the fabrication of the electrodes systems. Define the brand of the nitric acid and Rhodamine 6G (purchased/obtained from). Whether more feasible, provide for the missing info either in the main manuscript or in a separate SI document.
An English check would strongly boost the whole manuscript. Check for typos.
Moreover, it is suggested to list the abbreviations as a point-listed: it will boost the scientific appeal.
References list shall benefit additional broader state-of-the-art sources, in order to enhance the core discussion (e.g. J. Electroanal. Chem., 851 (2019) 113471).
The conclusion section would benefit further explanations, e.g. addition of a few sentences recapitulating the whole findings, the scientific progress and soundness. It might help discuss over an additional section any possible limitations/future perspectives.

Round 2

Reviewer 1 Report

It's a pity, but the paper still cannot be accepted in the present form due to the following reasons.

1) Extensive and careful editing of English language by the native speaker or professional editor is absolutely necessary.

2) Ligaments and pores are not the same; these are different elements of the microstructure.  However, authors use a mix of the both terms in the voluntary manner (See Fig. 1c and corresponding caption). It should be stated clearly in the Experimental how the size of pores and the size of ligaments were measured. What is aperture? Is it the same with pore or not?

3) Fig. 3 and corresponding caption. It is not clear what does single-atom layer REDOX reaction mean, what is surface-to-volume ratio, and how it was measured experimentally. In the lack of this information Fig. 3a is useless.

4) In general, comments to Fig.3 and Fig. 4 still don't promote the understanding of the reasons of enhanced SERS behavior of the sample processed at 1.05 V. 

Again, the paper should be reformatted to the short communication, without the extensive efforts to explain the observed phenomena. It is not done yet.

Reviewer 2 Report

Accept. Authors made the needed changes.

Round 3

Reviewer 1 Report

Several minor corrections of English are still desirable. However. in general, I am satisfied by revisions made by authors and, especially, by their comments.

The paper can be accepted in the present form.